# Enhancing Nature-Based Solutions: Efficient Removal of Hydroxytyrosol in Olive Mill Wastewater Treatment for Value Creation

Cecilia Faraloni [1], Eleftherios Touloupakis [2] and Eleonora Santos [3,*]

1   Istituto per la Bioeconomia, Consiglio Nazionale delle Ricerche (CNR), Via Madonna del Piano 10, 50019 Sesto Fiorentino, Italy; cecilia.faraloni@cnr.it
2   Istituto di Ricerca sugli Ecosistemi Terrestri, Consiglio Nazionale delle Ricerche (CNR), Via Madonna del Piano 10, 50019 Sesto Fiorentino, Italy
3   Centre of Applied Research in Management and Economics, Polytechnic Institute of Leiria, 2411-901 Leiria, Portugal
*   Correspondence: eleonora.santos@ipleiria.pt

**Abstract:** This study aimed to investigate the potential use of the microalgae *Chlorella sorokiniana* and *Scenedesmus quadricauda* for the bioremediation of olive mill wastewater (OMW), which is a major environmental issue and a waste product of olive oil production. The study investigated the effects of different dilutions (10% and 50%) of OMW on the growth of the microalgae and their ability to remove the phenolic component hydroxytyrosol (OH-Tyr) and enhance their antioxidant properties. The results indicated that, although the growth on OMW was not enhanced, both microalgae strains were able to remove OH-Tyr from OMW, with *Chlorella sorokiniana* showing higher removal efficiency than *Scenedesmus quadricauda*. Moreover, the antioxidant activity of the microalgal extracts increased after 96 h of exposure to OMW. These findings suggest that microalgae-based treatment of OMW could be a promising approach for the bioremediation of this waste product and the production of value-added products. Overall, the use of microalgae for the treatment of OMW could provide a sustainable solution for the management of this waste product while generating potential economic benefits for olive producers.

**Keywords:** microalgae-based treatments; nature-based solutions; olive mill wastewaters; economic sustainability

## 1. Introduction

Climate change is a significant factor that has led to an increase in the severity, frequency, and duration of droughts in many parts of Europe [1]. The economic and environmental impacts are significant, and there is a growing need for effective water resource management and drought mitigation solutions [2,3]. One promising approach to address these challenges is nature-based solutions (NBS), which can help restore soil biodiversity and sequester carbon. Compared to solely technological approaches, NBS are often more cost-effective options in the long run [4]. However, despite the potential of NBS for drought resilience, there is limited evidence of their cost-effectiveness and implementation in river basin management plans [4].

In recent years, microalgae-based treatments have emerged to be a sustainable and cost-effective method for treating olive mill wastewater (OMW) [5]. Due to the high content of organic matter and polyphenols, low pH, and dark color, the treatment of OMW is challenging. Disposal of OMW can be expensive and can also cause microbial soil alteration that affects soil structure and fertility [5]. However, phenolic compounds, particularly hydroxytyrosol, which are abundant in OMW, have been found to act as very strong antioxidants. This has led researchers to investigate the use of microalgae as an alternative to mechanical and chemical treatments for OMW [5].

Two microalgal strains, *Chlorella sorokiniana* and *Scenedesmus quadricauda*, were inoculated into OMW to evaluate their growth and phenolic acid recovery [4]. The OMW was previously centrifuged and diluted to 10% and 50% with BG11 medium. Although the growth on OMW was not enhanced, cells were able to remove hydroxytyrosol from the medium within 96 h. The hydroxytyrosol concentration in the cultures was reduced at both dilution rates by 47–50% (OMW 10%) and 60–70% (OMW 50%) for *Scenedesmus* and *Chlorella*, respectively [4]. The biomass of both strains was enriched with this phenolic acid, and microalgal extracts from the biomass of both strains showed a consistent increase in antioxidant properties, which were more than 50% higher than the initial values for both strains [6]. These results are of great significance both from an environmental and economic point of view, since hydroxytyrosol can find applications in the pharmaceutical and cosmetic industries, and the recovery of these compounds from OMW confers a significant added value for this wastewater [6].

The economic analysis of the technological application of the present strategy for polyphenol recovery could be an important next step [6]. The cost-effectiveness ratio of this approach needs to be evaluated in comparison with the traditional methods for the treatment of OMW [6]. The findings of this study could serve as a decision-support tool for olive oil producers looking for sustainable and cost-effective ways to manage their wastewater. It is also important to explore the potential of microalgae-based treatment methods for other types of wastewater and evaluate their potential for broader environmental and economic benefits [6,7].

The valorization of wastewater has been the subject of several studies and has been analyzed using different approaches. OMW has itself been considered a tool for removing toxic compounds, such as cesium and cobalt, from other wastewater [8].

Numerous studies have been conducted on the treatment of OMW using various methods. The most common methods for the treatment of OMW include physicochemical treatments, biological treatments, and advanced oxidation processes [9–28]. Among these methods, the use of microalgae for OMW treatment has been proposed as a sustainable and cost-effective approach [12,14,16,17,20–23,25,28].

In addition to the removal of pollutants, the use of microalgae for the treatment of OMW can also lead to the production of valuable biomass and bioactive compounds [14,16,17,20,22,25,28,29]. For instance, Elumalai et al. [29] investigated the use of the microalga *Scenedesmus quadricauda* for the simultaneous treatment of OMW and the production of bioactive compounds. The study found that the microalga was able to produce up to 1.24 g/L of biomass and up to 5.81 mg $L^{-1}$ of chlorophyll a. Another study by Ghernaout et al. [30] investigated the use of *Chlorella pyrenoidosa* for OMW treatment and biodiesel production. The study found that the microalga was able to remove up to 74% of COD and produce up to 17.67 g $L^{-1}$ of biomass.

However, there are still challenges that need to be addressed when treating OMW with microalgae. One of the major challenges is the low biomass productivity of microalgae grown in OMW, which can limit their potential for large-scale applications [14,17,22,25]. Tavakoli et al. [31] investigated the use of a hybrid microalgae-bacteria system for the treatment of OMW and found that the addition of bacteria can improve the biomass productivity of microalgae. Another challenge is the high concentration of inhibitory compounds, such as phenolic compounds, in OMW, which can negatively affect the growth and productivity of microalgae [12,14,17,22].

Overall, the literature suggests that microalgae treatment can be an effective and sustainable approach for treating OMW, leading to the removal of pollutants and the production of valuable biomass and bioactive compounds. However, further research is needed to optimize the growth and productivity of microalgae in OMW and to address the challenges associated with inhibitory compounds. The findings of this study on the use of nature-based solutions for the treatment of OMW contribute to the growing body of research on sustainable and eco-friendly approaches to wastewater treatment.

The aim of this paper is to investigate the potential use of microalgae, specifically *Chlorella sorokiniana* and *Scenedesmus quadricauda*, for the bioremediation of olive mill wastewater and the removal of the phenolic component hydroxytyrosol (OH-Tyr). The novelty aspects of this study include the following: (1) Evaluation of Growth and Hydroxytyrosol Removal: The study explores the effects of different dilutions of OMW on the growth of microalgae and their ability to remove hydroxytyrosol. It investigates the potential of these microalgae strains to effectively treat OMW, which is a major environmental issue and waste product of olive oil production; (2) Comparison of Microalgae Strains: The study compares the performance of *Chlorella sorokiniana* and *Scenedesmus quadricauda* in terms of hydroxytyrosol removal from OMW. It highlights the differences in removal efficiency between the two strains, with *Chlorella sorokiniana* showing higher removal efficiency; (3) Enhancement of Antioxidant Properties: The study examines the antioxidant activity of microalgal extracts after exposure to OMW. It reveals that the antioxidant activity of the microalgal extracts increases after 96 h of exposure, indicating the potential for value-added product generation; and (4) Potential Economic Benefits: The findings suggest that microalgae-based treatment of OMW could offer a promising approach for the bioremediation of this waste product while generating potential economic benefits for olive producers. The recovery of hydroxytyrosol from OMW adds significant value to the wastewater, making it a more sustainable and cost-effective solution.

These novel aspects contribute to the existing literature by expanding our understanding of microalgae-based treatment methods for OMW, highlighting their potential for hydroxytyrosol removal, and emphasizing the economic and environmental benefits of such approaches.

## 2. Material and Methods

### 2.1. Microalgal Growth

For the inoculum, the strains of *Chlorella sorokiniana* and *Scenedesmus quadricauda*, provided by IBE, CNR, Italy, were grown in tubular reactors with a working volume of 400 mL [32]. Cultures were grown on BG11 at a temperature of 28 °C and a light intensity of 100 μmol photons $m^{-2}$ $s^{-1}$, bubbled with air supplemented with $CO_2$, 98/2, *v/v*. In the late exponential growth phase, the cells of the cultures of both strains were harvested by centrifugation at $4000 \times g$ for 10 min and inoculated into sterilized OMW, which was previously filtered through a filter paper to remove the particulate component and diluted 10% and 50% with BG11 medium. The cultures were exposed to 400 μmol photons $m^{-2}$ $s^{-1}$ from one side at 25 °C. The composition of the OMW is reported in Table 1 [33].

**Table 1.** Chemical composition of olive mill wastewater (OMW) used for *Chlorella* and *Scenedesmus* cultured medium.

| OMW Composition | | |
|---|---|---|
| Total polyphenols | g $L^{-1}$ | 4.25 ± 0.15 |
| Phosphates | mg $L^{-1}$ | 672.00 ± 24.22 |
| Sulfates | mg $L^{-1}$ | 65.48 ± 11.45 |
| Nitrates | mg $L^{-1}$ | 8.70 ± 0.41 |
| Cu | mg $L^{-1}$ | 4.05 ± 1.21 |
| Mg | mg $L^{-1}$ | 195 ± 32.31 |
| K | mg $L^{-1}$ | 13.9 ± 2.73 |
| Ca | mg $L^{-1}$ | 635.00 ± 29.52 |
| Carbohydrates | g $L^{-1}$ | 9.01 ± 1.42 |
| Proteins | g $L^{-1}$ | 10.56 ± 1.79 |
| Ashes (%) | g $L^{-1}$ | 0.73 ± 0.04 |
| COD | g $O_2$ $L^{-1}$ | 50.90 ± 0.65 |
| BOD$_5$ | g $O_2$ $L^{-1}$ | 17.21 ± 1.47 |

The initial culture density was different for the two OMW concentrations, maintaining the same ratio of polyphenols concentration and biomass content. At 10% OMW concentration, the total chlorophyll (Chl a + Chl b) concentration was $7.5 \pm 0.50$ mg mL$^{-1}$ and $8.05 \pm 1.10$ mg mL$^{-1}$, corresponding to a biomass concentration of $602 \pm 12.5$ mg L$^{-1}$ and $650 \pm 55$ mg L$^{-1}$, for *C. sorokiniana* and *S. quadricauda*, respectively. At 50% OMW, the total chlorophyll concentration was $33.34 \pm 2.61$ mg mL$^{-1}$ and $35.82 \pm 1.10$ mg mL$^{-1}$, corresponding to a biomass concentration of $2650 \pm 100$ mg L$^{-1}$ and $2662 \pm 29$ mg L$^{-1}$, for *C. sorokiniana* and *S. quadricauda*, respectively. Cultures were maintained in batches under these conditions for 96 h. The same inoculum was used for each strain for the 10% and 50% OMW dilutions [32].

Pigments were extracted by vigorously agitating 100 mg of dried biomass with 5 mL of 90% acetone by vortexing for 5 min to extract the pigments. The supernatant was then removed by centrifugation at $4000 \times g$ for 5 min and transferred to another tube. The remaining pellet was subjected to this routine at least two more times until the pellet became colorless. The volume was set to 25 mL in a graduated flask with a 25 mL capacity. Dry weight was determined by filtering a known volume of the culture on a glass fiber disk, with a porosity of 2 μm, and washing the cells on the filter once with distilled water. Pigment analysis was performed by spectrophotometric determination of Chl a, Chl b, and total carotenoids in 90% acetone extracts, as previously described [34]. The analysis was performed in triplicate [32].

For phenolic compounds extraction, 100 mg of dried biomass was mixed with 10 mL of 100% methanol and sonicated for one minute at maximum power using a probe inserted into the liquid suspension. During sonication, ice was placed into the tubes containing the samples. In a graduated flask with a 25 mL capacity, the volume was set to 25 mL. All extracts were washed by adding petrol $1/1\ v/v$ to remove the interference from carotenoids and chlorophylls and then centrifuged at $4000 \times g$ for 5 min to recover the methanol fraction.

### 2.2. Phenolic Determination of Biomass Extracts

The methanol extracts produced as previously stated were analyzed with HPLC-DAD as described by Romani [35] for the identification and quantification of the various phenolic components. Sigma-HPLC Aldrich's standards were used for the identification and quantification of polyphenols. Analyses were carried out using a Varian (Palo Alto, CA, USA) ProStar 210 multisolvent pump and ProStar 335 photodiode array detector. A Phenomenex (Torrance, CA, USA) Kinetex Phenyl-Hexyl 100 A 150 × 4.6 mm reverse-phase C18 column with the same pre-column was used for the separation and analysis at 25 °C. (A) Water/acetic acid (99.9:0.1) and (B) methanol/water CH$_3$COOH made up the eluent (95:4.9:0.1). At a flow rate of 1.0 mL min$^{-1}$, a three-step linear solvent gradient system was used, commencing at 5% of solution B. From 2 to 22 min, the percentage of solution B was 25%; from 23 to 55 min, it was 99%; from 55 to 69 min, it was 5%. Chromatograms were recorded at 278 nm, while UV-Vis spectra were recorded in the 220–700 nm range. The analysis was carried out three times.

### 2.3. Antioxidant Properties of Microalgal Extracts

The free radical scavenging capacity of the microalgal polyphenolic extracts was determined in triplicate using the (2,2-diphenyl-1-picrylhydrazyl) (DPPH) reagent as described by Brand-Williams et al. [36].

A hydrogen-donor chemical that is antiradical can diminish the stable radical DPPH (Sigma-Aldrich, St. Louis, MO, USA). A Varian UV-Visible spectrophotometer Cary 50 Scan was used to quantify this colorimetric reaction at 517 nm. When the DPPH radicals reacted with the sample, the color of the radicals changed from violet to yellow. The absorbance at 517 nm of 1 mL of freshly prepared methanolic DPPH solution (63 M) was measured after adding 1 mL of adequately diluted methanolic extracts. After 20 min, the absorbance was measured a second time. To determine a curve and the method for calculating I50, or the concentration at which the initial DPPH absorbance is reduced by 50%, at least four

different sample concentrations were examined. The absorbance value at 517 nm of 1 mL of each sample at the same dilution of the analysis, added to 1 mL of methanol, was deducted from all the absorbance readings for each sample due to the green color of the extracts. For each extract, the IC50 was calculated using the following formula [36]:

$$\% \text{ inhibition} = [100 - (\text{Ax}/\text{As})] \times 100$$

where As is the initial absorbance of the sample extract in DPPH solution (t = 0) and Ax is the absorbance of the same sample after 20 min.

Antioxidant activity was measured as oxygen radical absorbance capacity (ORAC) assay according to Cao and Prior (1998) using a fluorescence spectrophotometer (Varian Cary Eclipse, Palo Alto, CA, USA). The sample was added to a free-radical generator (AAPH, 2,2′-azobis(2-aminopropane) dihydrochloride), and free radical inhibition was measured. Fluorescein was used as a target for free radical attacks. Fluorescence was measured every 5 min at excitation $\lambda$ 490 nm and emission $\lambda$ 512 nm. One ORAC unit refers to the value of the area under the curve (AUC) from the decay of fluorescence values to a stable value in the time considered (at least 30 min). The total relative ORAC value of the sample is reported as ORAC unit ($\mu$M) TE (Trolox equivalents) per mg of sample, given by the following formula:

$$\text{ORAC value} = k \, (\text{AUC}_{\text{sample}} - \text{AUC}_{\text{blank}})/(\text{AUC}_{\text{Trolox}} - \text{AUC}_{\text{blank}}) \times [\text{Trolox}]/[\text{Sample}], \tag{1}$$

where k is the dilution factor; AUC is area under the curve of the sample, blank, and Trolox, respectively; [Trolox] and [Sample] are the Trolox concentration (1 $\mu$mol) and the sample concentration (in mg).

The analysis was performed in triplicate.

### 2.4. Chlorophyll a Fluorescence Transient

The chlorophyll fluorescence transients were recorded on 15 min dark-adapted cells using a Handy-PEA (Hansatech Instruments, Pentney, UK) on 2 mL of liquid culture placed into a cell and illuminated with continuous light (650 nm peak wavelength, 3500 $\mu$mol photons m$^{-2}$ s$^{-1}$ light intensity) provided by light-emitting diodes (LEDs). Each chlorophyll a fluorescence-induction curve was analyzed using BiolyzerHP3 software following the so-called JIP-test [37].

Chlorophyll fluorescence data were normalized for both $F_0$ and $F_m$, with the transient calculated as relative variable fluorescence $V_t = (F_t - F_0)/(F_m - F_0)$, to facilitate comparison between samples [36].

The following parameters (JIP-test parameters), calculated from the fluorescence measurements, were considered: $M_o= 4(F300 \, \mu s - F_0)/(F_m - F_0)$, corresponding to the net rate of the reaction center closure; $V_J = (F_J - F_0)/(F_m - F_0)$, indicating the level of $Q_A$ reduction. The parameters describing the flux ratio were calculated according to Strasser et al. [38] as follows: $F_v/F_m$ is the maximum quantum yield of PSII for primary photochemistry; $\varphi E_o = F_v/F_m \cdot Y_o$ is the quantum yield of electron transport.

## 3. Results
### 3.1. Microalgal Growth

During contact with OMW, growth was affected in both strains at both dilutions. In particular, after 96 h, chlorophyll content decreased by 7% and 50% in both *Chlorella* and *Scenedesmus* at 10% and 50% OMW, respectively. These data suggest a different behavior and that 50% OMW, in particular, strongly reduces growth in terms of chlorophyll. Looking at the results of the dry weight measurements, it can be seen that the biomass content is also reduced (Figure 1). In addition to the 50% dilution, a significant decrease in dry weight was also observed at the 10% dilution. Specifically, for *Chlorella* and *Scenedesmus* exposed to 10% OMW, the dry weight decreased by 53% and 47%, respectively. At the higher OMW

concentration of 50%, dry weight decreased even further, by 64% and 58% for *Chlorella* and *Scenedesmus*, respectively.

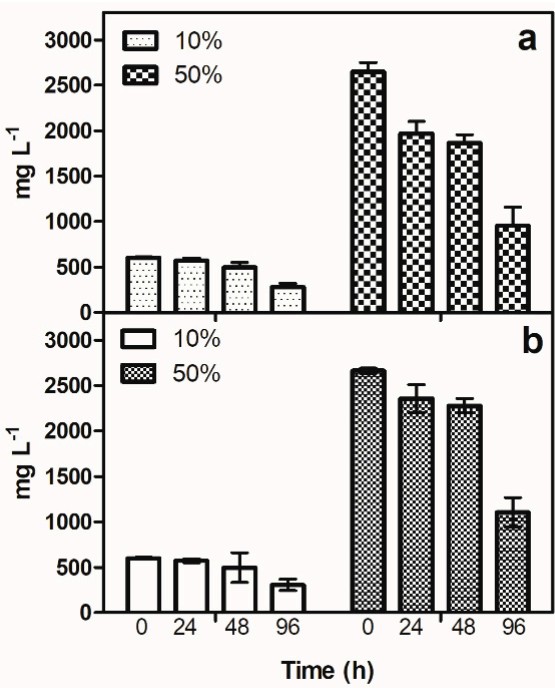

**Figure 1.** Effect of OMW dilution on dry weight biomass of *Chlorella* (**a**) and *Scenedesmus* (**b**). Bars indicate the standard deviation from three different experiments.

Figure 1 shows a significant decrease in dry weight biomass of both *Chlorella* and *Scenedesmus* strains at 10% and 50% OMW dilution compared to the control. The decrease in dry weight biomass was more pronounced for both strains at 50% OMW. These results suggest that OMW dilution has a negative impact on microalgal growth and biomass production.

### 3.2. Fluorescence Parameters

$V_J$ is a parameter that reflects the accumulation of electrons at the plastoquinone level and is associated with a reduction in photosynthetic activity and a decrease in $F_v/F_m$. When photosynthetic cells are under stress, they are unable to use all the light energy supplied and accumulate electrons at the photosynthetic apparatus, mainly at the plastoquinone level. The increase in $V_J$ measures this accumulation.

Table 2 shows the data on the changes in JIP test parameters during exposure to 10% OMW.

**Table 2.** Changes in JIP test parameters during exposure to 10% OMW. Each value represents the average of three different experiments (±SD).

|  | $F_v/F_m$ | $V_J$ | $\varphi Eo$ | $M_o$ |
|---|---|---|---|---|
| *Chlorella* |  |  |  |  |
| T0 | 0.702 ± 0.020 | 0.298 ± 0.036 | 0.503 ± 0.046 | 0.450 ± 0.047 |
| 96 h | 0.626 ± 0.047 | 0.391 ± 0.009 | 0.417 ± 0.052 | 0.482 ± 0.091 |
| *Scenedesmus* |  |  |  |  |
| T0 | 0.659 ± 0.008 | 0.284 ± 0.064 | 0:455 ± 0.041 | 0.389 ± 0.078 |
| 96 h | 0.548 ± 0.076 | 0.350 ± 0.036 | 0.362 ± 0.044 | 0.588 ± 0.041 |

The OMW had a negative impact on the photosynthetic activity of both *Chlorella* and *Scenedesmus* at this dilution, with a reduction in $F_v/F_m$ of only 11% and 17% for *Chlorella*

and *Scenedesmus*, respectively. An increase in $V_J$ by 31% and 19%, respectively, indicates electron accumulation at the level of the photosynthetic apparatus, which is also reflected in the reduction in electron transport rate indicated by the $\varphi E_o$ value, which was reduced by 17% and 20% for *Chlorella* and *Scenedesmus*, respectively. The functionality of the reaction centers was minimally affected in *Chlorella*, with an increase in the $M_o$ parameter of only 7%, whereas it was considerably altered in Scenedesmus with an increase of 51%.

　　　Table 3 shows the changes in JIP test parameters after 96 h of exposure to 50% OMW. The effects of OMW were severe, with significant changes in all parameters for both *Chlorella* and *Scenedesmus*. $F_v/F_m$ values decreased by 84%, while $V_J$ almost doubled. The $\varphi Eo$ values decreased 6.6-fold and 8.0-fold for *Chlorella* and *Scenedesmus*, respectively, indicating a significant reduction in electron transport rate. $M_o$ values were 2-fold and 3-fold higher than the initial values in *Chlorella* and *Scenedesmus*, respectively, indicating considerable alterations in the functionality of reaction centers.

**Table 3.** Changes in JIP test parameters of *Chlorella* and *Scenedesmus* after 96 h of exposure to 50% OMW. Each value represents the average of three different experiments (±SD).

|  | $F_v/F_m$ | $V_J$ | $\varphi E_o$ | $M_o$ |
|---|---|---|---|---|
| *Chlorella* |  |  |  |  |
| T0 | $0.702 \pm 0.020$ | $0.298 \pm 0.036$ | $0.503 \pm 0.046$ | $0.450 \pm 0.047$ |
| 96 h | $0.110 \pm 0.067$ | $0.598 \pm 0.098$ | $0.076 \pm 0.012$ | $0.929 \pm 0.127$ |
| *Scenedesmus* |  |  |  |  |
| T0 | $0.659 \pm 0.008$ | $0.284 \pm 0.064$ | $0.455 \pm 0.041$ | $0.389 \pm 0.078$ |
| 96 h | $0.106 \pm 0.015$ | $0.573 \pm 0.081$ | $0.057 \pm 0.014$ | $0.995 \pm 0.120$ |

*3.3. Hydroxytyrosol Removal from Culture Medium*

　　　The analysis of OMW revealed that hydroxytyrosol (OH-Tyr) was the predominant phenolic component, while other compounds were not detectable at the given dilution rates of 10% and 50%. The results of the phenolic removal study, which focused only on OH-Tyr, showed that its concentration in the culture medium of *C. sorokiniana* and *S. quadricauda* decreased within 96 h at both dilutions.

　　　Figure 2 shows the changes in OH-Tyr concentration in the cultures of the two strains at different dilutions and time points. At a dilution of 10%, a slight decrease in OH-Tyr concentration was observed after 24 h only in *C. sorokiniana*, but after 48 h the reduction was evident in both strains.

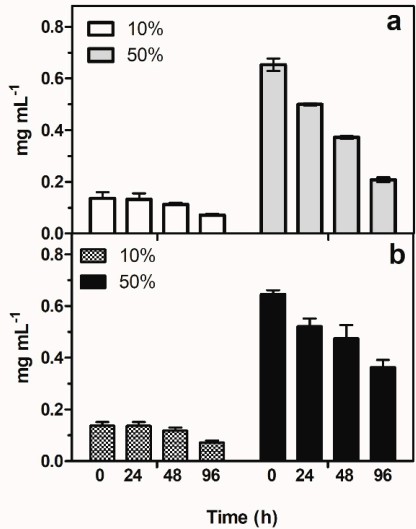

**Figure 2.** Changes in hydroxytyrosol concentration in OMW cultures of *C. sorokiniana* (**a**) and *S. quadricauda* (**b**) at different dilutions and time points. Bars indicate the standard deviation from three different experiments.

At 50% dilution, a higher reduction was observed in *Chlorella* than in *Scenedesmus* cultures, and the decrease in OH-Tyr concentration was significant after 24 h. This trend of OH-Tyr removal was consistent throughout the experiment, with the highest removal efficiency observed in *Chlorella* cultures.

Figure 3 shows that the trend of OH-Tyr removal based on biomass content is consistent with the previous results. After 96 h, *Chlorella* exhibited 12% higher total hydroxytyrosol removal compared to *Scenedesmus* at 10% dilution. At 50% dilution, the difference in biomass removal was even more pronounced, with *Chlorella* exhibiting a 36% higher removal rate than *Scenedesmus*.

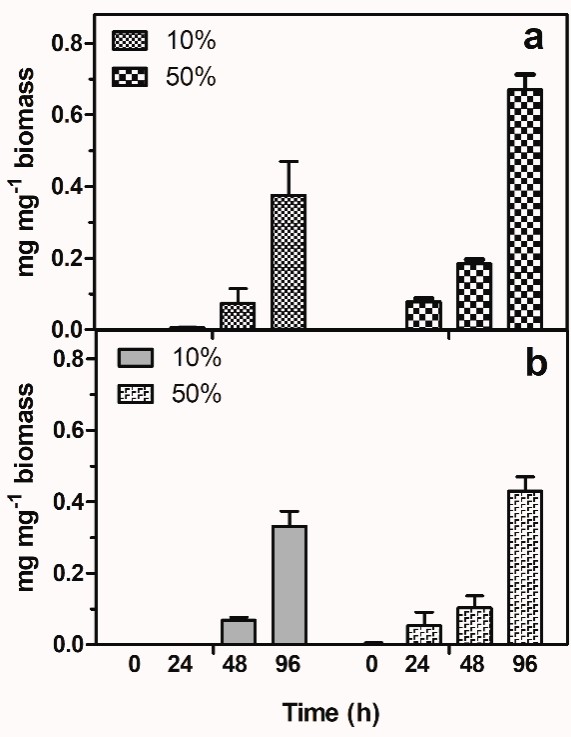

**Figure 3.** Comparison of OH-Tyr removal by biomass content in *Chlorella* (**a**) and *Scenedesmus* (**b**) cultures at different dilution rates of OMW. Bars indicate the standard deviation from three different experiments.

### 3.4. Antioxidant Activity

The results of the measurements of the antiradical activity of the biomass extracts are reported in Table 4. As observed for both strains and both percentages of OMW, the $I_{50}$ decreased, after 96 h. For *Chlorella* extracts, $I_{50}$ decreased by 62% and 82%, whereas in *Scenedesmus* extracts, it decreased by 39% and 77%, with OMW 10% and 50%, respectively.

**Table 4.** Antiradical activity of the biomass extracts in the culture medium in *Chlorella* and *Scenedesmus* with 10% and 50% OMW. Each value represents the average of three different experiments ($\pm$SD).

|  | 10% | 50% |
|---|---|---|
|  | $I_{50}$ | $I_{50}$ |
| *Chlorella* | $\mu g\ mL^{-1}$ | $\mu g\ mL^{-1}$ |
| 0 | 74.7 $\pm$ 11.7 | 74.7 $\pm$ 11.7 |
| 96 h | 28.2 $\pm$ 5.2 | 13.3 $\pm$ 0.3 |
| *Scenedesmus* |  |  |
| 0 | 89.2 $\pm$ 10.8 | 89.2 $\pm$ 10.8 |
| 96 h | 54.4 $\pm$ 2.8 | 20.2 $\pm$ 0.9 |
| Gallic acid | 2.01 $\pm$ 0.60 |  |

The decrease in $I_{50}$ followed the trend of OH-Tyr removal in both strains. In both strains, the accumulation of OH-Tyr in the biomass corresponded to the decrease in I50, indicating the important role of this phenolic compound in conferring antioxidant properties to the microalgal cells.

This trend was also evident in the antioxidant activity measured with ORAC. The results are reported in Table 5. The results indicate that the antioxidant activity in the biomass extracts of the cultures grown on the different percentages of OMW increases sharply in both strains. In *Chlorella*, ORAC activity increased 2-fold and 6-fold, at 10% and 50% OMW, respectively. In *Scenedesmus*, it increased by 41% and 5.3-fold, at 10% and 50% OMW, respectively. In addition, in this case, the highest OH-Tyr accumulation corresponded to the highest ORAC activity.

**Table 5.** Antioxidant activity of the biomass extracts in the culture medium in Chlorella and Scenedesmus with 10% and 50% OMW. Each value represents the average of three different experiments ($\pm$SD).

|  | 10% | 50% |
|---|---|---|
| *Chlorella* | $\mu$molTE mg$^{-1}$ | $\mu$molTE mg$^{-1}$ |
| 0 | $0.051 \pm 0.006$ | $0.051 \pm 0.006$ |
| 96 h | $0.101 \pm 0.012$ | $0.295 \pm 0.012$ |
| *Scenedesmus* | | |
| 0 | $0.044 \pm 0.005$ | $0.044 \pm 0.005$ |
| 96 h | $0.062 \pm 0.006$ | $0.232 \pm 0.017$ |
| Gallic acid | $26.5 \pm 2.4$ | |

The differences between $I_{50}$ and ORAC activity of the two strains were less pronounced at 50% OMW, although an appreciable/significant difference OH-Tyr accumulation was detected. Indeed, OH-Tyr removal was higher in *Chlorella* cultures than in *Scenedesmus* cultures.

## 4. Discussion

This study aimed to investigate the potential use of microalgae, specifically *Chlorella sorokiniana* and *Scenedesmus quadricauda*, for the bioremediation of olive mill wastewater (OMW) and the removal of the phenolic compound hydroxytyrosol (OH-Tyr). The findings of this study contribute to the existing literature by providing deeper insights into the performance of microalgae in OMW treatment and highlighting their potential for hydroxytyrosol removal.

The results showed that both *Chlorella sorokiniana* and *Scenedesmus quadricauda* were able to grow in the presence of OMW and effectively remove hydroxytyrosol, the main phenolic compound in OMW. This aligns with previous studies that have demonstrated the ability of microalgae to treat wastewater and accumulate bioactive compounds. Huang et al. [38] reported the effective removal of nutrients and organic matter from wastewater by the microalgae Scenedesmus obliquus, while Ghernaout et al. [30] found that microalgae can remove organic pollutants and heavy metals while producing valuable biomass. Elumalai et al. [29] observed a significant reduction in total phenolic content and increased biomass antioxidant activity in Chlorella vulgaris exposed to tannery effluent. Furthermore, Tavakoli et al. [31] demonstrated the ability of Scenedesmus sp. to remove phenolic compounds from synthetic wastewater and produce bioactive compounds with antioxidant properties.

In comparison to previous studies, one notable feature of the present experiment was the use of real OMW instead of synthetic wastewater, which better mimics the real conditions in the olive mill industry. Additionally, two different strains of microalgae were employed to compare their growth and removal efficiency. The findings revealed that while both strains were capable of hydroxytyrosol removal, *Chlorella sorokiniana* exhibited higher removal efficiency.

Moreover, the fluorescence parameters showed that both microalgae strains experienced physiological stress when exposed to OMW, with a dose-dependent effect observed. The presence of higher OMW concentrations, such as 50%, led to a decrease in light penetration and hindered efficient light harvesting, resulting in a decrease in photosynthetic efficiency and inhibition of growth. Although the growth of microalgae was not sustained in the presence of OMW, the removal of hydroxytyrosol remained high and comparable to other studies, indicating the potential of microalgae strains for effective hydroxytyrosol removal without the need for additional costly and time-consuming treatments.

Furthermore, the antioxidant activity of the microalgae was evaluated, a desirable trait for potential commercial applications. The findings corroborate previous studies that have demonstrated the enhancement of antioxidant properties in microalgae after exposure to OMW. This suggests the potential for obtaining enriched polyphenolic biomass using OMW and opens new perspectives for commercial applications in the food and cosmetic industries.

However, it is essential to acknowledge the limitations of this study. Firstly, it focused on the removal of hydroxytyrosol and did not analyze other phenolic compounds present in OMW, which could influence the overall treatment efficiency. Secondly, the investigation was limited to two strains of microalgae, and it is possible that other species may exhibit different responses to OMW exposure. Therefore, future studies should consider evaluating a wider range of microalgae strains to gain a more comprehensive understanding of their performance. Additionally, the potential toxicity of microalgae after exposure to OMW was not assessed, which is crucial when considering their use as biofertilizers or animal feed.

In a nutshell, the findings of this study demonstrate the potential of microalgae, particularly *Chlorella sorokiniana* and *Scenedesmus quadricauda*, for the treatment of OMW and the removal of hydroxytyrosol. The use of microalgae in OMW treatment offers a sustainable and cost-effective solution to address the environmental challenges posed by this waste product in olive oil production. Furthermore, the production of bioactive compounds with antioxidant properties presents potential economic benefits for olive producers. However, further investigations are necessary to comprehensively evaluate the treatment efficiency, analyze the presence of other phenolic compounds, assess the toxicity of microalgae, and determine the economic feasibility and potential risks associated with their application in agricultural and industrial sectors.

## 5. Conclusions

The results reported here provide important insights into the potential of microalgae to treat OMW and generate added value through the production of bioactive compounds. Our results show that the microalgae strains *Chlorella sorokiniana* and *Scenedesmus quadricauda* are able to efficiently remove hydroxytyrosol, the most abundant phenolic compound in olive mill wastewater, while simultaneously increasing their antioxidant activity.

Nonetheless, some limitations should be considered when interpreting the results. First, the performance of only two microalgal strains was investigated, and it is possible that other strains have different or even better treatment efficiencies. Further studies are needed to comprehensively assess the potential of microalgal-based treatment of olive mill wastewater.

Despite these limitations, important implications may be envisaged with the microalgae-based treatment of olive mill wastewaters for olive producers, the environment, and potential profits. By treating olive mill wastewater with microalgae, producers can achieve environmental compliance and reduce the negative impacts of wastewater discharge on soil and water resources. Moreover, after the treatment, the microalgal biomass can be used as a source of bioactive compounds, other than hydroxytyrosol, such as antioxidants, pigments, and lipids, with potential applications in the food, pharmaceutical, and cosmetic industries. This could create new opportunities for sustainable and profitable utilization of olive mill wastewater while contributing to the circular economy.

Further research is needed to fully exploit the potential of this innovative approach and its application in the real-world context.

**Author Contributions:** Conceptualization, E.S. and C.F.; methodology, E.S. and C.F.; software, C.F. validation, E.S. and C.F.; formal analysis, E.S., C.F. and E.T.; investigation, writing—original draft preparation, E.S., C.F. and E.T; writing—review and editing, E.S., C.F. and E.T.; supervision, E.S. and C.F.; project administration E.S.; funding acquisition, E.S. All authors have read and agreed to the published version of the manuscript.

**Funding:** This research is financed by National Funds of the FCT—Portuguese Foundation for Science and Technology within the project «UIDB/04928/2020» and, under the Scientific Employment Stimulus-Institutional Call CEECINST/00051/2018.

**Institutional Review Board Statement:** Not applicable.

**Informed Consent Statement:** Not applicable.

**Data Availability Statement:** Not applicable.

**Conflicts of Interest:** The authors declare no conflict of interest.

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
