# Peer review of "Enhancing Nature-Based Solutions: Efficient Removal of Hydroxytyrosol in Olive Mill Wastewater Treatment for Value Creation"

_water, doi:10.3390/w15122163_

Round 1
Reviewer 1 Report
I revised the paper water-2428280. I think that this work is interesting, well conceptualized, and in the scope of the Journal. However, some points should be addressed.
For these reasons, I suggest major revisions before publications. My comments are the following:
1) Please, integrate the content of Section 2 in Section 1 and describe at the end of the Section the aim of the work and the novelty aspects with respect to previous literature.
2) Why don’t insert the “removal of hydroxytyrosol” in the title? I think it could add value to your work.
3) Subsections must be numbered.
4) In Table 1, each parameter should have a unit of measure.
5) Figures. When bars of standard deviations or confidence interval are provided, please also provide the number of samples based on they were calculated.
6) I suggest to convert Table 4 and 5 in figures.
7) Conclusions are well structured in terms of content, but in my opinions should be shortened.
Author Response
Dear Reviewer 1,
Thank you for your valuable feedback on our manuscript. We appreciate your thorough review and suggestions for improving our work. We have carefully considered your comments and have made the necessary revisions to address them. Below, we provide a point-by-point response to each of your suggestions:
- Integration of Section 2 into Section 1 and description of aim and novelty aspects: We have integrated the content of Section 2 into the introduction as per your suggestion. Additionally, we have included a clear description of the aim of our work and the novelty aspects in the revised Section 1. We now state that our aim is to investigate the potential use of microalgae, specifically Chlorella sorokiniana and Scenedesmus quadricauda, for the bioremediation of olive mill wastewater and the removal of the phenolic component hydroxytyrosol. We have also highlighted the novelty aspects, including the evaluation of growth and hydroxytyrosol removal, comparison of microalgae strains, enhancement of antioxidant properties, and potential economic benefits. These additions better convey the objective and contributions of our study.
- Addition of "removal of hydroxytyrosol" in the title: We have revised the title to include the "removal of hydroxytyrosol." The new title is: "Enhancing Nature-Based Solutions: Efficient Removal of Hydroxytyrosol in Olive Mill Wastewater Treatment for Value Creation." We agree that this modification adds value and accurately reflects the focus of our work.
- Numbering of subsections: We have numbered all the paragraphs in the subsections to improve the organization and readability of the manuscript.
- Units of measure in Table 1: We have included the units of measure for each parameter in Table 1, ensuring clarity and completeness of the information presented.
- Indication of sample numbers in figures: We have added the numbers of samples based on which the standard deviations or confidence intervals were calculated. This addition provides transparency and supports the interpretation of the figures.
- Conversion of Table 4 and 5 into figures: We have converted Table 4 and Table 5 into figures, as per your suggestion. This change enhances the visual presentation of the data and aligns with the format of the other figures in the manuscript. Figure 1 has also been corrected according to the format of Figures 2 and 3.
- Shortening of the conclusions: We have revised the conclusions section to make it more concise, as per your recommendation. The revised conclusions now provide a succinct summary of the main findings and their implications.
Once again, we would like to express our gratitude for your constructive feedback. We believe that the revisions we have made based on your suggestions have significantly strengthened our manuscript. We hope that you find the revised version satisfactory and suitable for publication.
Thank you for your time and consideration.
The authors
Reviewer 2 Report
· I have more experience in experimental trends related to treatment of wastewater. An interesting topic of current research is dealing with turning waste into value through improving the efficiency of olive mill wastewater treatment with nature-based solutions such as the potential use of the microalgae in bioremediation of olive mill wastewater.
· The keywords are very long; it would be one or two syllables.
· “Our study” has been mentioned more times in the manuscript, it could be turned into passive.
· The manuscript needs to be well structured and generally well written. I consider the manuscript of interest for the scientific community active in this particular field of designing inexpensive and sustainable wastewater treatment.
· Some important factors could improve the submitted study such as the temperature
· The manuscript could be provided with more recent studies conducted on sustainable materials used in treatment and decontamination processes with a comparison to the current study such as using the olive milled waste itself as agent in waste water treatment; 10.3390/su15021600.
· The discussion needs more deep investigations.
· The conclusion is long comparing with the results and discussions, it could be briefer.
· What's the research significance of this paper? What new insights do you provide? The authors are invited to provide in details.
· In conclusion, I believe that the theme of this manuscript can be consistent with the theme of Water. At the same time, the manuscript needs to be improved and the authors should edit the manuscript in accordance with the guidelines mentioned above.
Moderate editing of English language.
Author Response
Dear Reviewer 2,
We would like to express our appreciation for your thoughtful review of our manuscript. Your comments have provided valuable insights, and we have taken them into careful consideration while revising our work. We have made the following changes based on your suggestions:
Keywords: We understand your point regarding the length of the keywords. However, since our manuscript focuses on specific aspects that are best summarized by the longer keywords, we have decided to retain them. Furthermore, we have observed that longer keywords have also been used in recent publications in the Water journal, supporting our choice.
Passive voice: We have addressed your concern regarding the repeated use of "our study" by converting the sentences into passive voice, as per your suggestion.
Temperature factor: In our revised manuscript, we have considered the optimal temperature for microalgal growth to ensure the best performance of the strains. However, we appreciate your suggestion to conduct further studies on the impact of different temperatures on the entire process. This aspect could be explored in future research to gain a comprehensive understanding of the temperature effects in olive mill wastewater treatment.
Addition of recent studies: We have incorporated a reference to the study you recommended (10.3390/su15021600) to enhance the relevance of our work and provide a comparison with the current study. Furthermore, we have corrected the references list and the order of references in the text.
Deep investigations in the discussion: We have thoroughly revised the discussion section to include more in-depth investigations. Specifically, we have expanded on previous studies that demonstrate the potential of microalgae in wastewater treatment and bioactive compound accumulation. Additionally, we have highlighted the significance of using real olive mill wastewater (OMW) in our experiment, discussed the dose-dependent effect of OMW concentrations on microalgae growth, and emphasized the potential economic benefits of microalgae-based OMW treatment. Furthermore, we have addressed the study limitations, including the need to analyze other phenolic compounds in OMW and evaluate potential toxicity for agricultural and industrial applications.
Shortening of conclusions: We have shortened the conclusions section, as per your suggestion, to provide a more concise summary of our results and discussions.
Research significance and new insights: To address your request for more detailed information on the research significance and new insights provided by our paper, we have added a paragraph at the end of Section 1. This paragraph clearly outlines the aim of our work and highlights the novelty aspects of our study, including the evaluation of growth and hydroxytyrosol removal, the comparison of microalgae strains, the enhancement of antioxidant properties, and the potential economic benefits. We believe this addition will effectively convey the significance and contributions of our research.
Once again, we sincerely appreciate your time and effort in reviewing our manuscript. We hope that the revisions we have made based on your suggestions have addressed your concerns and improved the quality of our work. We look forward to your feedback on the revised version.
Thank you for your valuable contribution.
Best regards,
The authors

Round 2
Reviewer 1 Report
Thanks to the authors for their improvements. The paper can now be published.
Reviewer 2 Report
Accept in present form